# Comparison of Two Kinds of Two-Dimensional Shear Wave Elastography Techniques in the Evaluation of Jaundiced Infants Suspected of Biliary Atresia

**DOI:** 10.3390/diagnostics12051092

**Published:** 2022-04-27

**Authors:** Wenying Zhou, Jinyu Liang, Quanyuan Shan, Huadong Chen, Pengfei Gao, Qinghua Cao, Guotao Wang, Xiaoyan Xie, Luyao Zhou

**Affiliations:** 1Department of Medical Ultrasonics, Institute of Diagnostic and Interventional Ultrasound, The First Affiliated Hospital of Sun Yat-sen University, Sun Yat-sen University, Guangzhou 510080, China; zhouwy6@mail2.sysu.edu.cn (W.Z.); ljyu@mail.sysu.edu.cn (J.L.); shanqy3@mail.sysu.edu.cn (Q.S.); wanggt6@mail2.sysu.edu.cn (G.W.); 2Department of Pediatric Surgery, The First Affiliated Hospital of Sun Yat-sen University, Sun Yat-sen University, Guangzhou 510120, China; chenhd25@mail2.sysu.edu.cn (H.C.); gaopf3@mail.sysu.edu.cn (P.G.); 3Department of Pathology, The First Affiliated Hospital of Sun Yat-sen University, Sun Yat-sen University, Guangzhou 510080, China; caoqhua@mail.sysu.edu.cn

**Keywords:** biliary atresia, ultrasonography, elasticity imaging techniques, liver fibrosis

## Abstract

Purpose: To compare the reliability and performance of Supersonic shear wave elastography (S-SWE) and Toshiba shear wave elastography (T-SWE) in the diagnosis of biliary atresia (BA) and assessment of liver fibrosis among jaundiced infants suspected of BA. Material and Methods: A total of 35 patients with suspected BA who underwent both S-SWE and T-SWE examinations were prospectively included. Diagnostic performances of S-SWE and T-SWE in identifying BA were evaluated. The correlation between two types of SWE values and histological liver fibrosis stages by Metavir scores were investigated in 21 patients with pathology results. The intraclass correlation coefficients (ICCs) were calculated in 16 patients for inter- and intra-observer agreement. The area under the receiver operating characteristic curve (AUC) analysis was compared using a DeLong test. Results: There were 22 patients with BA and 13 patients without BA. The diagnostic performance of S-SWE was comparable to that of T-SWE (AUC 0.895 vs. 0.822, *p* = 0.071) in diagnosing BA. The AUCs of S-SWE in predicting liver fibrosis stages were from 0.676 to 1.000 and showed no statistical differences from that of T-SWE (from 0.704 to 1.000, all *p* > 0.05). T-SWE provided higher inter-operator agreement (ICC 0.990) and intra-operator agreement (ICCs 0.966–0.993), compared with that of S-SWE in a previous study (ICC 0.980 for inter-operator and 0.930–0.960 for intra-operator). Conclusions: For infants suspected of BA, T-SWE had good performances in the diagnosis of BA and the assessment of liver fibrosis compared with S-SWE. Furthermore, T-SWE showed higher measurement reproducibility than S-SWE.

## 1. Introduction

Biliary atresia (BA) is a severe cholestatic disease presenting in infancy with established obliteration of the biliary tree [1]. The assessment of liver stiffness, which is well-correlated with liver fibrosis, is important for the diagnosis and the prognostic estimation of patients with BA [2,3,4,5,6]. A variety of elastography techniques have been developed to assess the degree of liver fibrosis [3,7,8,9,10] and to help identify BA in infants with conjugated hyperbilirubinemia [11,12,13,14,15,16,17,18]. However, the diagnostic performance of elastography varied a lot among various ultrasound (US) machines from different corporations in the diagnosis of BA, with areas under the receiver operating characteristic (ROC) curve (AUCs) from 0.790 to 0.997 [11,12,13,14,15,16,17,18]. Thus, the elastography technique of which US machine is the best for jaundiced infants is still confusing clinicians.

Transient elastography is a quantitative elastography technique that has been used to diagnose BA and evaluate liver fibrosis in pediatric patients [11,16,19], but the inability to choose different locations for the region of interest limits its application [20]. Two-dimensional shear wave elastography (2D-SWE) is based on capturing shear wave speed propagation, presenting a map of the elasticity in one area and allowing stiffness quantitative analysis [4]. Currently, Supersonic shear wave elastography (S-SWE; SuperSonic Imagine, Aix-en-Provence, France) is one of the most widely used 2D-SWE technologies in the diagnosis of BA [2,13,18] and the assessment of liver fibrosis [3,21,22]. But there is no objective evaluation for image quality. Another 2D-SWE technique, Toshiba SWE (T-SWE; Canon Medical System, Tochigi, Japan), has parameters to evaluate image stability. However, there was only one publication describing the diagnosis of BA with T-SWE [12]. Until now, it remains unknown whether the parameters of image stability are helpful in the evaluation of patients with BA.

Thus, this study aimed to compare the reliability and performance of S-SWE and T-SWE in the diagnosis of BA and the assessment of liver fibrosis for jaundiced infants suspected of BA.

## 2. Materials and Methods

### 2.1. Patient Enrollment

This prospective study was approved by the ethics committee of the First Affiliated Hospital of Sun Yat-sen University, and written informed parental consent was obtained.

From November 2015 to May 2016, a total of 48 patients with conjugated hyperbilirubinemia were prospectively evaluated. The inclusion criteria were as follows: (1) conjugated hyperbilirubinemia of unknown cause; (2) successful liver stiffness measurements by both S-SWE and T-SWE; (3) the final diagnosis was clear. Patients who did not fully meet the inclusion criteria were excluded. As part of routine clinical care, serum biochemical tests were carried out for all patients within 3 days of the SWE examination. Data including total bilirubin (TB), direct bilirubin (DB), alanine aminotransferase (ALT) and aspartate aminotransferase (AST) were obtained.

Finally, 35 patients were enrolled, including 22 infants with BA and 13 infants without BA. Of the 22 infants with BA, 13 were confirmed by surgical exploration, 7 were confirmed by intraoperative cholangiography under laparoscopy and 2 were confirmed by liver biopsy. Of the 13 infants without BA, 4 were confirmed by cholescintigraphy, 6 were confirmed by liver biopsy and 3 were confirmed by intraoperative cholangiography under laparoscopy.

### 2.2. The Measurement of S-SWE

S-SWE examinations were performed in all patients by a single radiologist (with 10 years of experience in ultrasound and 4 years of experience in elastography) with an Aixplorer scanner (SuperSonic Imagine, Aix-en-Provence, France), incorporating a linear array transducer SL15-4 with the frequency ranging from 5 to 14 MHz. All infants were kept quiet by feeding when measuring. Ultrasonic elastography measurements were performed on segment V or VI of the liver. The transducer was kept perpendicular to the skin, avoiding vessels, bile ducts, or artefacts from rib or lung gas, without applying any pressure. The size of the S-SWE rectangular region of interest was 3.0 cm × 2.0 cm with a circular region of interest (0.7–1.2 cm diameter) in the center. The scan procedure was the same as previously reported [13].

### 2.3. The Measurement of T-SWE

After the S-SWE examination was completed, T-SWE examinations were performed on the same segment by the same operator as above. The US scanner was Aplio500 (Canon Medical System, Otawara, Tochigi, Japan) and incorporated a liner array transducer 14-L5 with the frequency ranging from 5 to 14 MHz. The preparation of the patient was the same as that mentioned for the S-SWE. The elasticity mode of T-SWE was selected for the comparison with S-SWE. The size of the T-SWE rectangular region of interest was 1.8 cm ×1.5 cm with a circular region of interest (0.5–1.0 cm diameter) in the center.

For both S-SWE and T-SWE examinations, three liver stiffness measurements were performed on each infant and the mean value was recorded for subsequent statistical analyses.

### 2.4. Liver Histopathology

Among 35 infants with suspected BA, 21 had pathology results, of which 13 were obtained by surgical exploration and 8 by ultrasound-guided percutaneous liver biopsy. Liver fibrosis was evaluated by one pathologist (with 10 years of experience in liver pathology) in these 21 infants based on the Metavir classification as follows [23]: F0, no fibrosis; F1, portal fibrosis with no septa; F2, portal fibrosis with rare fibrous septa; F3, bridging fibrosis with many fibrous septa; and F4, cirrhosis. Among them, the severity of liver fibrosis was divided into three groups: ≥F2 (significant fibrosis), ≥F3 (advanced fibrosis) and F4 (cirrhosis) [24].

### 2.5. Inter-Operator and Intra-Operator Error of T-SWE and S-SWE

When assessing the inter- and intra-operator error between these two types of elastography, 16 patients were randomly selected to undergo T-SWE measurements operated by two additional radiologists (both radiologists with 6 years of experience in ultrasound and 2 years of experience in elastography), and the results of the T-SWE were compared with those of the S-SWE reported in the previous study [13]. Of the 16 patients, 10 had BA and 6 did not have BA. All the measurements were performed as described above. Each radiologist performed three measurements and each measurement was performed on the same liver segment. The mean values of the three measurements between the two radiologists were compared to assess the inter-operator error. Three measurements per radiologist were obtained to determine the intra-operator error.

### 2.6. Statistical Analysis

Continuous variables were first tested for normality by using a Kolmogorov–Smirnov test. Normally distributed variables were expressed as mean ± standard deviation (SD), whereas non-normally distributed variables were expressed as median and interquartile range (IQR). Comparisons between groups were made with an unpaired *t*-test for normally distributed variables or a Mann–Whitney U test for skewed variables. The chi-squared test was used to assess gender differences between BA and non-BA groups. The Spearman’s rank coefficient test was performed to assess the correlation between variables. AUCs were compared by a DeLong test to evaluate the diagnostic performances of S-SWE and T-SWE [25]. The optimal cut-off values were determined based on the largest Youden index. The inter- and intra-operator errors of T-SWE were tested by the intraclass correlation coefficient (ICC). The statistical analysis was performed with the SPSS 20.0 software (SPSS, Chicago, IL, USA) and MedCalc Statistical Software version 15.2.2 (MedCalc Software bvba, Ostend, Belgium). All statistical tests were two sided, and *p* < 0.05 indicated significant difference.

## 3. Results

### 3.1. Patient Characteristics

The clinical characteristics between infants with BA and without BA are displayed in Table 1. No significant differences were observed in terms of age, sex or the levels of ALT and AST (all *p* > 0.05), but there were differences in the levels of TB (*p* = 0.013) and DB (*p* = 0.003) between the two groups. The values of S-SWE and T-SWE were significantly higher in the BA group (Figure 1) compared to the non-BA group (Figure 2) (*p* < 0.001 and 0.002, respectively). In terms of liver fibrosis stages, there was F0 in one patient, F1 in five, F2 in six, F3 in seven and F4 in two. The distributions of the values of T-SWE and S-SWE for each Metavir fibrosis stage are shown in Figure 3. The distributions of liver stiffness values in histological liver stages ≥F2 were not statistically different between the two elastography techniques (all *p >* 0.05).

### 3.2. Diagnostic Performances of S-SWE and T-SWE

The diagnostic performances of S-SWE and T-SWE in the diagnosis of BA and the prediction of liver fibrosis are summarized in Table 2. According to the ROC curve, the optimal cutoff value of T-SWE was 8.7 kPa for the diagnosis of BA. The optimal cut-off value for identifying BA by S-SWE was 10.2 kPa, calculated from the previous study [13]. S-SWE had a higher AUC in the diagnosis of BA (0.895, 95% CI 0.745–0.973) than that of T-SWE (0.822, 95% CI 0.655–0.930), but the difference was not statistically significant (*p* = 0.071) (Figure 4).

On the contrary, T-SWE had a higher AUC than that of S-SWE in predicting significant fibrosis (0.811 vs. 0.706), and advanced fibrosis (0.704 vs. 0.676) with the corresponding cut-off values of 8.9 vs. 13.0 kPa and 13.5 vs. 14.0 kPa, respectively (Table 2). However, the differences between S-SWE and T-SWE in predicting significant fibrosis and advanced fibrosis were also not statistically significant (*p* = 0.211 and 0.619, respectively).

### 3.3. The Correlation between 2D-SWE Values and Serum Biochemical Tests

Correlation analysis showed that the levels of TB, DB, ALT and AST were significantly correlated with the values of T-SWE (r ranged from 0.368 to 0.588, all *p* < 0.05). On the other hand, except for ALT (r = 0.298 and *p* = 0.103), the above indexes were also significantly correlated with S-SWE (r ranged from 0.430 to 0.546, *p* < 0.05). The stiffness values of the two elastography techniques had the highest correlation with the levels of DB (r = 0.546 for S-SWE and 0.588 for T-SWE), and had the lowest correlation with the levels of ALT (r = 0.298 for S-SWE and 0.368 for T-SWE).

### 3.4. Inter- and Intra-Operator Error

Sixteen patients were included in the assessment of the inter-operator and intra-operator error of T-SWE, for the comparison with that of the S-SWE reported in the previous study [13]. As previously reported, the ICC of the S-SWE was 0.980 (95% CI -0.960–0.990) for inter-operator, whereas the ICCs were ranged from 0.930 to 0.960 for intra-operator (95% CI 0.860–0.980). When measuring by T-SWE the 16 patients in this study, the ICC of inter-operator was 0.990 (95% CI 0.993–0.977) and the ICCs of intra-operator were ranged from 0.966 (95% CI 0.926–0.987) to 0.993 (95% CI 0.983–0.997).

## 4. Discussion

Our study revealed that both S-SWE and T-SWE examinations had a promising diagnostic performance in identifying BA among infants with conjugated hyperbilirubinemia (AUC 0.895 for S-SWE and 0.822 for T-SWE). We also demonstrated that both elastography techniques had good diagnostic performances in assessing the liver fibrosis stages. Furthermore, good intra- and inter-operator consistency of T-SWE was observed in this study, which indicates that T-SWE is robust enough for the assessment of liver stiffness with high reproducibility. Importantly, the T-SWE technique showed higher measurement reproducibility than S-SWE (ICC 0.990 for T-SWE and ICC 0.980 for S-SWE of inter-operator). It is probably because T-SWE provides the propagation map and speed smart map with more visible regional shear wave propagation characteristics for evaluation [26]. Therefore, radiologists can evaluate the single-shot acquisitions better. Furthermore, it also helps to optimize the region of interest positioning, allowing operators to avoid regions with heterogeneous propagation characteristics.

In the present study, we demonstrated that the liver stiffness value of patients with BA is higher than that with non-BA, which is similar to previous studies [13,14,15,18]. Our previous study reported an AUC of 0.790 (95% CI 0.659–0.826) in predicting BA by S-SWE in 172 infants with conjugated hyperbilirubinemia [13], which was slightly worse than the present research (AUC 0.895, 95% CI 0.745–0.973). This may be due to sample bias. In this study, the AUC of T-SWE in diagnosing BA was 0.822, which was comparable with that of S-SWE (0.895) (*p* = 0.071). These results indicated that both T-SWE and S-SWE had good performances in identifying BA from infants with obstructive jaundice.

The study also showed that S-SWE and T-SWE had similar diagnostic performances in predicting different fibrosis stages. However, the cutoff values for each liver fibrosis stage varied between the two elastography techniques, which implies that it is not suitable to directly compare the results of liver stiffness measurements from different scanners. When evaluating liver fibrosis using S-SWE equipped with SL15-4 probes, the thresholds for different liver fibrosis stages ranged from 7.3 to 32.8 kPa in the previous study [21]. In this study, the thresholds of S-SWE for the prediction of significant fibrosis, advanced fibrosis and cirrhosis were 13.0, 14.0 and 27.3 kPa, respectively, which were similar with the ranges demonstrated [21]. However, the thresholds of T-SWE for the prediction of significant fibrosis, advanced fibrosis and cirrhosis were 8.9, 13.5 and 14.7 kPa, respectively, which were all lower than that of S-SWE. Therefore, the same elastography technique should be used when patients are in need of following up on their liver fibrosis stages.

Previous studies have shown that the stiffness values could be affected by intrahepatic cholestasis, which was closely related to the level of bilirubin [27,28]. A similar result is also observed in this study. The stiffness values of both elastography techniques had the highest correlation with the levels of DB. Guo et al. [28] found that intrahepatic cholestasis showed a slight effect on the LSM value from S-SWE, also leading to overestimation of liver fibrosis stages. In our study, the mean S-SWE value was higher than the mean T-SWE value in 86.4% of BA patients, which may be due to the fact that S-SWE is more likely to be affected by cholestasis than T-SWE. Furthermore, it was reported that liver stiffness values would be increased after food intake, especially in adult patients with cirrhosis and portal hypertension [29,30,31], but the increase usually peaks 20~40 min after food intake and only adds 1 kPa at the peak [30]. The liver stiffness measurements in this study were all completed within 20 min of feeding, so feeding may have a potential but insignificant effect on the liver stiffness values for both systems.

Several limitations could be found in our study. Firstly, this study only compared the diagnostic performance of two types of 2D-SWE techniques in the diagnosis of BA and the evaluation of liver fibrosis. Other different 2D-SWE systems can be used for comparison in further study. Secondly, the number of patients was small, which may cause a potential impact on statistical validity. Further studies are needed in the future to expand the sample size to further explore the impact of different machine parameter settings on the study results.

In conclusion, both S-SWE and T-SWE were feasible and stable in the diagnosis of BA and the assessment of liver fibrosis. However, the T-SWE technique might have higher measurement reproducibility than S-SWE.

## Figures and Tables

**Figure 1 diagnostics-12-01092-f001:**
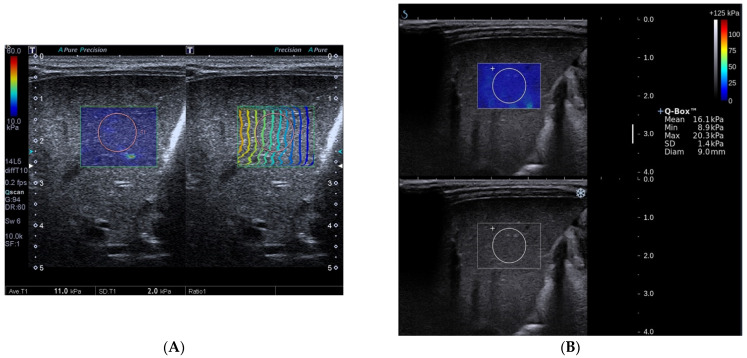
Toshiba shear wave elastography measurements (**A**) and Supersonic shear wave elastography measurements (**B**) in a 45-day-old boy. The final diagnosis of the patient was biliary atresia and Metavir fibrosis stage was F3. A, Speed smart map of Toshiba shear wave elastography (left) shows the mean liver stiffness value was 11.0 kPa. The propagation map of Toshiba shear wave elastography shows regularly parallel contour lines (right). B, Supersonic shear wave elastography measurement in the regions of interest (top, blue areas) shows the mean liver stiffness value was 16.1 kPa.

**Figure 2 diagnostics-12-01092-f002:**
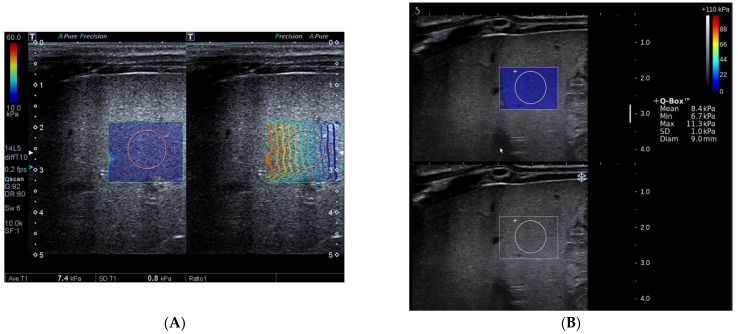
Toshiba shear wave elastography measurements (**A**) and Supersonic shear wave elastography measurements (**B**) in a 78-day-old boy. The final diagnosis of the patient was non-biliary atresia (transient cholestasis of unknown etiology) and Metavir fibrosis stage was F3. A, Speed smart map of Toshiba shear wave elastography (left) shows the mean liver stiffness value was 7.4 kPa. The propagation map of Toshiba shear wave elastography shows regularly parallel contour lines (right). B, Supersonic shear wave elastography measurement in the regions of interest (bottom, blue areas) shows the mean liver stiffness value was 8.4 kPa.

**Figure 3 diagnostics-12-01092-f003:**
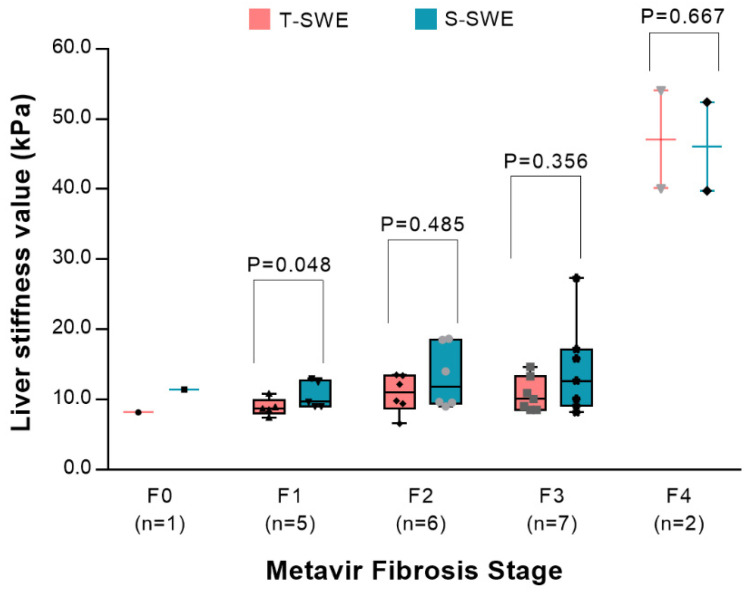
Box-and-whisker plots showing the distribution of Toshiba shear wave elastography values (red boxes) and Supersonic shear wave elastography values (blue boxes) for each Metavir fibrosis stage. Data are number of patients in brackets.

**Figure 4 diagnostics-12-01092-f004:**
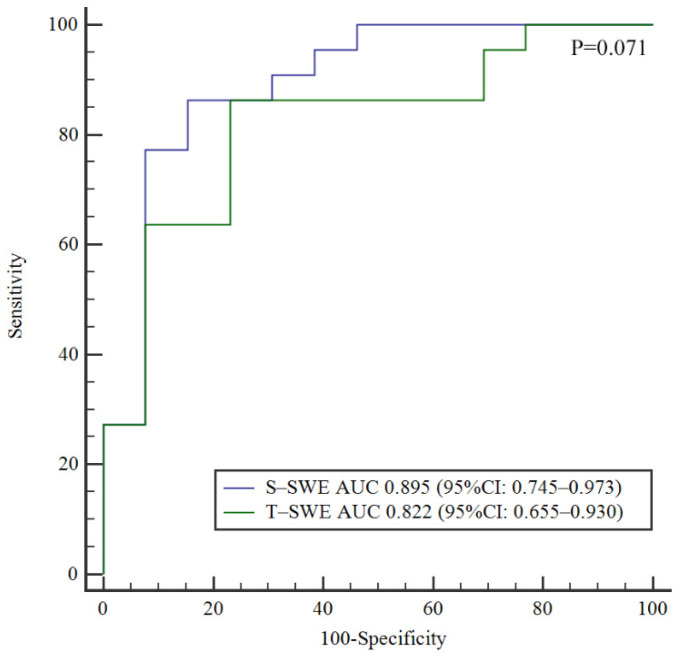
Comparison of ROC curves for Toshiba shear wave elastography and Supersonic shear wave elastography in the identification of biliary atresia.

**Table 1 diagnostics-12-01092-t001:** Baseline characteristics at time of shear wave elastography measurements in pediatric patients with conjugated hyperbilirubinemia.

Characteristics	BA (*n* = 22)	Non-BA (*n* = 13)	*p* Value
Age (days)	61 (45–75)	69 (50–87)	0.330
Male-to-female ratio	16:6	10:3	0.557
Total bilirubin level (μmol/L)	189.2 (159.1–273.9)	152.3 (108.1–188.9)	0.013
Direct bilirubin level (μmol/L)	102.3 (90.2–147.1)	86.1 (47.6–93.9)	0.003
Alanine aminotransferase level (U/L)	134.0 (84.0–175.0)	97.0 (49.0–163.0)	0.231
Aspartate aminotransferase level (U/L)	214.0 (171.0–331.0)	129.0 (79.0–334.0)	0.201
S-SWE value (kPa)	14.0 (11.1–20.0)	8.2 (7.1–9.7)	<0.001
T-SWE value (kPa)	11.0 (9.1–13.5)	8.5 (6.5–9.2)	0.002

Note: BA, biliary atresia; S-SWE, Supersonic shear wave elastography; T-SWE, Toshiba shear wave elastography. Except for the data in Male-to-female ratio, data are medians, with interquartile range in parentheses.

**Table 2 diagnostics-12-01092-t002:** Diagnostic performances of S-SWE and T-SWE in identifying biliary atresia among 35 patients and predicting liver fibrosis among 21 patients.

	BA (*n* = 22) vs. Non-BA (*n* = 13)	F0-1 (*n* = 6) vs. F ≥ 2 (*n* = 15)	F0-2 (*n* = 12) vs. F ≥ 3 (*n* = 9)	F0-3 (*n* = 19) vs. F = 4(*n* = 2)
S-SWE				
Cut-off value (kPa)	>10.2	>13.0	>14.0	>27.3
AUC	0.895 (0.745–0.973)	0.706 (0.469–0.881)	0.676 (0.439–0.861)	1.000 (0.839–1.000)
Sensitivity (%)	77.3 (54.2–91.3)	53.3 (27.4–77.7)	55.6 (22.7–84.7)	100.0 (19.8–100.0)
Specificity (%)	84.6 (53.7–97.3)	100.0 (51.7–100.0)	83.3 (50.9–97.1)	100.0 (79.1–100.0)
Accuracy (%)	80.0 (63.8–90.3)	66.7 (45.2–83.0)	71.4 (49.8–86.4)	100.0 (81.8–100.0)
PPV (%)	89.5 (65.5–98.2)	100.0 (59.8–100.0)	71.4 (30.3–94.9)	100.0 (19.8–100.0)
NPV (%)	68.8 (41.5–87.9)	46.2 (20.4–73.9)	71.4 (42.0–90.4)	100.0 (79.1–100.0)
T-SWE				
Cut-off value (kPa)	>8.7	>8.9	>13.5	>14.7
AUC	0.822 (0.655–0.930)	0.811 (0.583–0.946)	0.704 (0.467–0.880)	1.000 (0.839–1.000)
Sensitivity (%)	86.4 (64.0–96.4)	80.0 (51.4–94.7)	33.3 (9.0–69.1)	100.0 (19.8–100.0)
Specificity (%)	76.9 (46.0–93.8)	83.3 (36.5–99.1)	100.0 (69.9–100.0)	100.0 (79.1–100.0)
Accuracy (%)	82.9 (66.9–92.3)	81.0 (59.4–92.9)	71.4 (49.8–86.4)	100.0 (81.8–100.0)
PPV (%)	86.4 (64.0–96.4)	92.3 (62.1–99.6)	100.0 (31.0–100.0)	100.0 (19.8–100.0)
NPV (%)	76.9 (46.0–93.8)	62.5 (25.9–89.8)	66.7 (41.2–85.6)	100.0 (79.1–100.0)
*p* value *	0.071	0.211	0.619	1.000

Note: 95% confidence intervals are included in brackets. * The *p* values were from the comparison between the AUC of the S-SWE and that of T-SWE by a DeLong test. S-SWE, Supersonic shear wave elastography; T-SWE, Toshiba shear wave elastography; BA, biliary atresia; PPV, positive predictive value; NPV, negative predictive value; AUC, area under the receiver operating characteristic curve.

## Data Availability

Data can be requested from the corresponding authors upon reasonable request.

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
