# Peer review of "Comparison of Two Kinds of Two-Dimensional Shear Wave Elastography Techniques in the Evaluation of Jaundiced Infants Suspected of Biliary Atresia"

_diagnostics, 2022, doi:10.3390/diagnostics12051092_

Round 1

Reviewer 1 Report

The developed work explores the comparison between two diagnostic methods based on elastography to quantify its potential and difference in the evaluation of jaundiced children suspected of biliary atresia.

In my opinion, there are several methodological issues at the statistical level that would be specified and improved, despite being of an adequate level.

For example, the difference of means is not reflected as a procedure in terms of sample power (1), the chi square statistic must be specified more precisely for its decision (2) and the choice of the normality test is omitted (3). The difference of AUCs in terms of DeLong is not a standard and explicit procedure in SPSS it has a special interest to compare groups such as F0, .. F4, which are not represented graphically (4). The plots should be edited and a box-whiskers graphics should be edited including significance and outliers (5).

Despite concluding that there are no differences between this two techniques, there are indications of significance and the sample size is low, this should be mentioned (6) and I recommend at least a paragraph in the discussion describing an explanation about the correlation with other parameters (7).

Author Response

Dear editors and Reviewers,

Thank you for the very valuable comments and suggestions! We have revised the manuscript as suggested, with main revisions highlighted. The detailed response to each comment is listed below.

Reviewer 1:

The developed work explores the comparison between two diagnostic methods based on elastography to quantify its potential and difference in the evaluation of jaundiced children suspected of biliary atresia.

In my opinion, there are several methodological issues at the statistical level that would be specified and improved, despite being of an adequate level.

For example, the difference of means is not reflected as a procedure in terms of sample power (1), the chi square statistic must be specified more precisely for its decision (2) and the choice of the normality test is omitted (3).

Response: Thanks for your suggestion. We have revised the manuscript and added more details into the “Statistical analysis” section.

The difference of AUCs in terms of DeLong is not a standard and explicit procedure in SPSS it has a special interest to compare groups such as F0, .. F4, which are not represented graphically (4).

Response: Thanks for your comments. The AUCs were compared by a DeLong test using MedCalc Statistical Software version 15.2.2 in this study. The distributions of the two elasticity techniques at different fibrosis severity levels were compared using the Mann-Whitney U test, and P-values have been added in the revised Figure 3.

The plots should be edited and a box-whiskers graphics should be edited including significance and outliers (5).

Response: Thanks for your suggestion. We have revised Figure 3 based on your suggestion.

Despite concluding that there are no differences between this two techniques, there are indications of significance and the sample size is low, this should be mentioned (6)

Response: Thanks for your suggestion. We have added the comments about the limitations of small sample size affecting statistical power in the “Discussion” section.

and I recommend at least a paragraph in the discussion describing an explanation about the correlation with other parameters (7).

Response: Thanks for your suggestion. The correlation of liver stiffness values with indicators of liver function has been explored in this study, and its potential significance has been explored independently in the fourth paragraph of the Discussion.

Reviewer 2 Report

In this study the authors aimed to compare the reliability and performance of two different shear-wave techniques in the diagnosis of biliary atresia and in the assessment of liver fibrosis for jaundiced infants suspected of biliary atresia. As the authors specified in the limitation section, the number of patients was small, which may influence the statistical validity. However, this study provides important information on the use of 2d-swe techniques in the diagnosis and evaluation of patients with biliary atresia.

At Material and methods:

  1. The authors state “All infants were kept quiet 85 by feeding when measuring”, in adults Food intake increases liver stiffness in patients and is considered a confounding factor. Please add a comment about this fact.
  2. Which kind of Aixplorer device was used? Aixplorer Mach 30? Mach 20? Is Aixplorer from Paris, France? or from Aix-en-Provence, France?
  3. In the introduction chapter when you state that “ 2D-SWE is valuable for the assessment of liver fibrosis” please cite this paper :
    Popa A, Șirli R, Popescu A, Bâldea V, LupuÈ™oru R, Bende F, Cotrău R, Sporea I. Ultrasound-Based Quantification of Fibrosis and Steatosis with a New Software Considering Transient Elastography as Reference in Patients with Chronic Liver Diseases. Ultrasound Med Biol. 2021 Jul;47(7):1692-1703. doi: 10.1016/j.ultrasmedbio.2021.02.029. Epub 2021 Apr 6. PMID: 33832824.

Author Response

Dear editors and Reviewers,

Thank you for the very valuable comments and suggestions! We have revised the manuscript as suggested, with main revisions highlighted. The detailed response to each comment is listed below.

Reviewer 2:

In this study the authors aimed to compare the reliability and performance of two different shear-wave techniques in the diagnosis of biliary atresia and in the assessment of liver fibrosis for jaundiced infants suspected of biliary atresia. As the authors specified in the limitation section, the number of patients was small, which may influence the statistical validity. However, this study provides important information on the use of 2d-swe techniques in the diagnosis and evaluation of patients with biliary atresia.

At Material and methods:

  1. The authors state “All infants were kept quiet 85 by feeding when measuring”, in adults Food intake increases liver stiffness in patients and is considered a confounding factor. Please add a comment about this fact.

Response: OK. We have added the comments in the “Discussion” section.

  1. Which kind of Aixplorer device was used? Aixplorer Mach 30? Mach 20? Is Aixplorer from Paris, France? or from Aix-en-Provence, France?

Response: The device used in this study is Aixplorer, which is from Aix-en-Provence, France. We have revised this information in the “methodology” section.

  1. In the introduction chapter when you state that “ 2D-SWE is valuable for the assessment of liver fibrosis” please cite this paper :Popa A, Șirli R, Popescu A, Bâldea V, LupuÈ™oru R, Bende F, Cotrău R, Sporea I. Ultrasound-Based Quantification of Fibrosis and Steatosis with a New Software Considering Transient Elastography as Reference in Patients with Chronic Liver Diseases. Ultrasound Med Biol. 2021 Jul;47(7):1692-1703. doi: 10.1016/j.ultrasmedbio.2021.02.029. Epub 2021 Apr 6. PMID: 33832824.

Response: OK. We have cited this paper in the “introduction” chapter.

Round 2

Reviewer 1 Report

The manuscript seems to be be correct and the authors have changed the issues according to the suggestions. The references should be introduced in a correct format and I recommend a second checking of the language.